# Cysteine-specific protein multi-functionalization and disulfide bridging using 3-bromo-5-methylene pyrrolones

Yingqian Zhang [1,3], Chuanlong Zang[1,3], Guoce An[2], Mengdi Shang[1], Zenghui Cui[1], Gong Chen [1], Zhen Xi[1] & Chuanzheng Zhou [1✉]

Many reagents have been developed for cysteine-specific protein modification. However, few of them allow for multi-functionalization of a single Cys residue and disulfide bridging bioconjugation. Herein, we report 3-bromo-5-methylene pyrrolones (3Br-5MPs) as a simple, robust, and versatile class of reagents for cysteine-specific protein modification. These compounds can be facilely synthesized via a one-pot mild reaction and they show comparable tagging efficiency but higher cysteine specificity than the maleimide counterparts. The addition of cysteine to 3Br-5MPs generates conjugates that are amenable to secondary addition by another thiol or cysteine, making 3Br-5MPs valuable for multi-functionalization of a single cysteine and disulfide bridging bioconjugation. The labeling reaction and subsequent treatments are mild enough to produce stable and active protein conjugates for biological applications.

[1] State Key Laboratory of Elemento-Organic Chemistry and Department of Chemical Biology, College of Chemistry, Nankai University, Tianjin 300071, China. [2] Department of Forensic Chemistry, Criminal Investigation Police University of China, Liaoning, Shenyang 110854, China. [3] These authors contributed equally: Yingqian Zhang, Chuanlong Zang. ✉email: chuanzheng.zhou@nankai.edu.cn

Cysteine modification is a practical approach for producing functionalized proteins and has found widespread biomedical applications[1–3]. Unpaired, free cysteine, which generally exhibits high nucleophilicity, is relatively rare in proteins and is therefore an ideal site for chemoselective protein modification. Accessible free cysteine residues can be readily introduced by site-directed mutagenesis. Cysteine-specific modification is typically achieved by reaction of the thiol group with electrophiles such as maleimides, iodoacetamides, alkyl halides, and pyridyl disulfides[4,5]. Recently, the need to prepare homogeneous antibody–drug conjugates has inspired a wave of interest in exploring cysteine-specific bioconjugation reagents with improved reactivity, specificity, stability, and biocompatibility[6,7]. To this end, maleimide derivatives[8–10], dehydroalanine[11,12], perfluoroaromatic reagents[13–16], organometallic palladium reagents[17,18], and many others[19–23] have been developed, and each of these reagents shows unique merits. However, only a few of them permit multi-functionalization of a single Cys residue[22,24,25], which would be advantageous in certain situations.

In proteins, disulfide bonds are more common than free cysteines, and disulfide reduction followed by rebridging bioconjugation offers another approach for chemoselective modification of proteins and is especially attractive for functionalization of immunoglobulin G (IgG), an antibody that has four reducible interchain disulfides but no free cysteine[25–28]. Many reagents, including bromomaleimides[29,30], bromopyridazinediones[27,31–33], bis-sulfones and allyl sulfones[24,26,34], and divinylpyrimidine[28], have been developed for such applications. Nevertheless, seldom of them meet the requirements of simplicity, robustness, and high efficiency.

Recently, we reported that 5-methylene pyrrolones (5MPs) are promising protein-modification reagents[10]. These compounds are easy to prepare, stable, and highly thiol-specific. 5MP–protein conjugates can undergo traceless loss of the 5MP moiety when the pH is increased or when they are subjected to thiol-exchange conditions, thus making 5MPs valuable for reversible protein modification. Herein, we report that 3-bromo-5MPs (3Br-5MPs), derivatives of 5MPs, nevertheless exhibit different cysteine-specific protein bioconjugation behavior (Fig. 1). These brominated analogs retain merits of 5MPs such as easy preparation and high thiol-specificity but show improved reactivity toward cysteine. In addition, protein-3Br-5MP conjugates can undergo addition reactions with another thiol group, which makes 3Br-5MPs valuable tools for multi-functionalization of a single cysteine residue and disulfide bridging bioconjugation. The obtained adducts can be reduced under mild conditions to generate stable and biologically active conjugates. We demonstrate that 3Br-5MPs are simple and robust reagents for protein mono-, multi-functionalization, and disulfide bridging bioconjugation.

## Results

**Preparation of 3Br-5MPs.** 3Br-5MPs were first synthesized by means of the same strategy used for 5MPs[10,35]. Specifically, a variety of primary amine substrates (**2a–g**) were coupled with intermediate **1′** in neutral aqueous solution to furnish the corresponding 3Br-5MPs (**3a–g**) in modest yields (Method 1 in Table 1). We also prepared 3Br-5MPs from 4-bromo-furfuryl acetate (**4**) via a one-pot reaction. Oxidation of **4** by N-bromo-succinimide (NBS) gave the intermediate **1′** [36,37], which was in situ trapped by **2a–g** to provide the corresponding 3Br-5MPs (Method 2 in Table 1). Method 2 was more facile than method 1 and afforded 3Br-5MPs in higher yields.

We assessed the stability of the 3Br-5MPs by incubating **3a** under various conditions and found that it, similar to the 5MP counterpart **5** (Fig. 2), was completely stable over the course of a 5-day incubation period in a relatively wide pH range (6.0–9.5). In contrast, maleimide analog **6** decomposed rapidly under these conditions (Supplementary Figs. 22 and 23). Hence, 3Br-5MPs can be more easily prepared and are more stable than the widely used cysteine-specific bioconjugation reagents—maleimides.

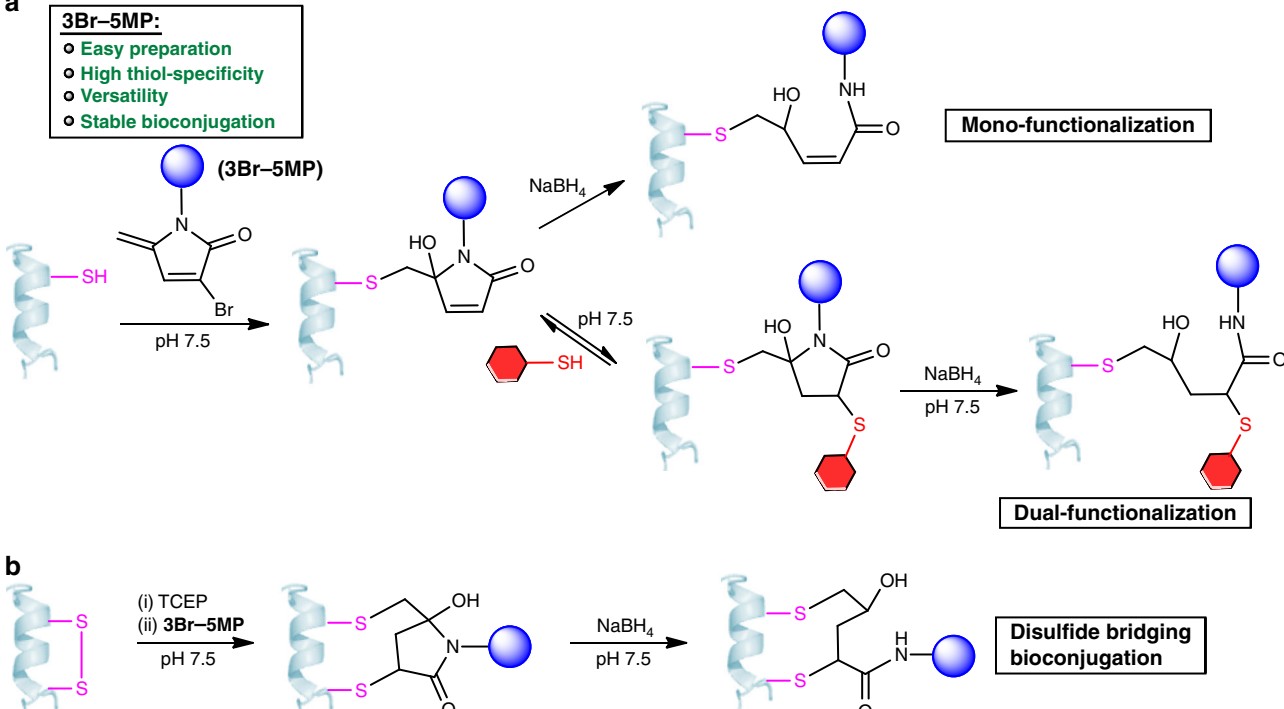

**Fig. 1 Use of 3-bromo-5-methylene pyrrolones (3Br-5MPs) as versatile reagents for cysteine-specific protein modification. a** Protein mono-functionalization and dual-functionalization using 3Br-5MP. **b** Disulfide bridging bioconjugation of protein using 3Br-5MP.

## Table 1 Yields of preparation of 3Br-5MPs by different methods.

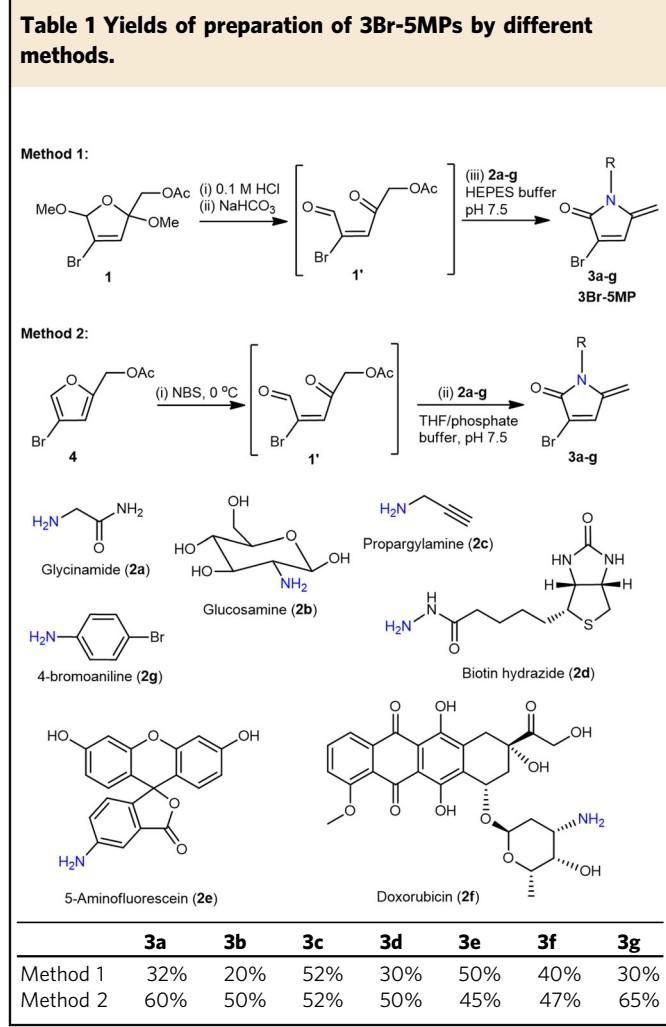

|  | 3a | 3b | 3c | 3d | 3e | 3f | 3g |
|---|---|---|---|---|---|---|---|
| Method 1 | 32% | 20% | 52% | 30% | 50% | 40% | 30% |
| Method 2 | 60% | 50% | 52% | 50% | 45% | 47% | 65% |

**Fig. 2 Analogues of 3Br-5MP used for comparison in this study.**
Structures of 5MP (**5**) and maleimide (**6**).

### Reactivity of 3Br-5MPs toward thiols.

The reactivity of the 3Br-5MPs toward thiol groups was evaluated with **3a** as a model compound. Treatment of **3a** with 5 equiv of EtSH in neutral aqueous solution at 37 °C for 5 min generated α, β-unsaturated lactam **7** as the sole product (Fig. 3a). To fully characterize the structure of EtSH-3Br-5MP adducts, **8** was prepared and its structure was determined by X-ray diffraction analysis (Fig. 3b). The *S*-ethyl moiety is attached at the C6 position, suggesting that just like 5MP, the addition of thiols to 3Br-5MPs occurs via 1, 6-addition. The loss of the C3-Br atom and appearance of the C5-OH group in compound **8** point to an addition–elimination mechanism.

Unlike thiol–maleimide adducts and thiol–5MP adducts, which decompose rapidly via hydrolysis and/or retro-Michael reactions[10,38], **7** was stable for up to 2 days in aqueous solution at pH 6.0–8.5 (Supplementary Fig. 24). However, we found **7** was reactive toward thiols. Treatment of **7** with excess EtSH (20 equiv) quantitatively generated dual-adducts **9** as a pair of

diastereoisomers in 30 min (Fig. 3c). This result suggests that addition of EtSH to **7** was considerably slower than the addition of EtSH to **3a** ($k_2 \ll k_1$); otherwise, a mixture of **7** and **9** would have been obtained in the latter case.

Where the second EtSH is anchored in **9** could not be determined by nuclear magnetic resonance (NMR) at this stage because of the complexity of the isomers. This could, however, be confirmed after their transformation to reduced forms (see below), which subsequently showed that the second EtSH was introduced at the C3 position. This suggests that a carbonyl function may be in situ generated at the C5 position during the addition of EtSH to the mono-adduct **7**. Taken together, we proposed the following mechanism for the reaction between thiols and 3Br-5MPs: the primary addition of one thiol to 3Br-5MP generates a highly electrophilic intermediate **10** (Fig. 3d), which is attacked by hydroxide, followed by elimination of the C3-Br to give a stable mono-adduct (**11**). Compound **11** itself, an α, β-unsaturated amide, is not a good Michael acceptor[39], but it can isomerize to a ring-open form (**12**). In the intermediate **12**, Michael addition of thiol to the C3 position is supposed to be much favorable than an addition to the C4 position, given that α, β-unsaturated ketones are much better Michael acceptors than α, β-unsaturated amides[39,40]. Therefore, Michael addition of another thiol to the C3 position of **12** gives an enolate intermediate **12′**, which is subjected to isomerization and a subsequent intramolecular ring-close reaction, providing the dual-adduct **13**. Given that mono-adduct **7** was complete stable after incubation in neutral buffers for 48 h and no ring-open products were detected (Supplementary Fig. 24), isomerization of **11** to the ring-open form **12** is likely unfavorable and thus **12** exists in a trace amount. This indicates that ring-open is the rate-limiting step and explains, at least partly, why the secondary addition ($k_2$) is much slower than the primary one ($k_1$).

We found the dual-adduct **9** was liable to thiol-exchange reaction. Treating **9** with 20 equiv. of β-mercaptoethanol (BME) gave **14** slowly (Fig. 4a). This thiol-exchange reaction suggests that the dual-adduct **9** is not stable and the Michael addition of thiols to the mono-adduct **7** is reversible. That is, isomerization of **9** to the open-ring form (**15**) leads to the elimination of EtSH via a retro-Michael reaction to give the intermediate **16**. Michael addition of BME to **16** generates the observed product **14** (Fig. 4b). In this regard, we proposed that reduction of the C5-ketone in **15** to hydroxyl may significantly retard the elimination reaction and thus transfer the dual-adduct to a stabilized form. Thus, **9** was treated with 1.3 equiv. of NaBH4 in pH 7.5 buffer and it was quantitatively reduced to **17** in minutes (Fig. 4c). The major isomer of **17** was isolated and fully characterized by NMR, which revealed that the second EtSH was introduced at the C3 position (Supplementary Figs. 16–18). Compound **17** was stable and no decomposition was observed after incubation in pH 7.5 buffer for 48 h (Supplementary Fig. 25). Similarly, treatment of **7** with NaBH4 resulted in the ring-opening and reduced mono-adducts **18**, which was stable and not active toward BME anymore (Fig. 4d and Supplementary Fig. 26).

### Reactivity of 3Br-5MPs upon reaction with peptides and proteins.

Next, we investigated the reactivity and cysteine specificity of 3Br-5MPs by carrying out reactions with peptide **19**, which contains only one cysteine, in the middle of the sequence (Table 2). Treatment of peptide **19** with 2 equiv. of **3a** in HEPES buffer (pH 7.5) for 5 min afforded cysteine-modified products in more than 96% yield at both 37 °C and 4 °C (Supplementary Fig. 27). The cysteine-tagging efficiency of **3a** was clearly higher than that of 5MP analog **5** and was comparable to that of maleimide analog **6** (Table 2). To compare the cysteine specificity of

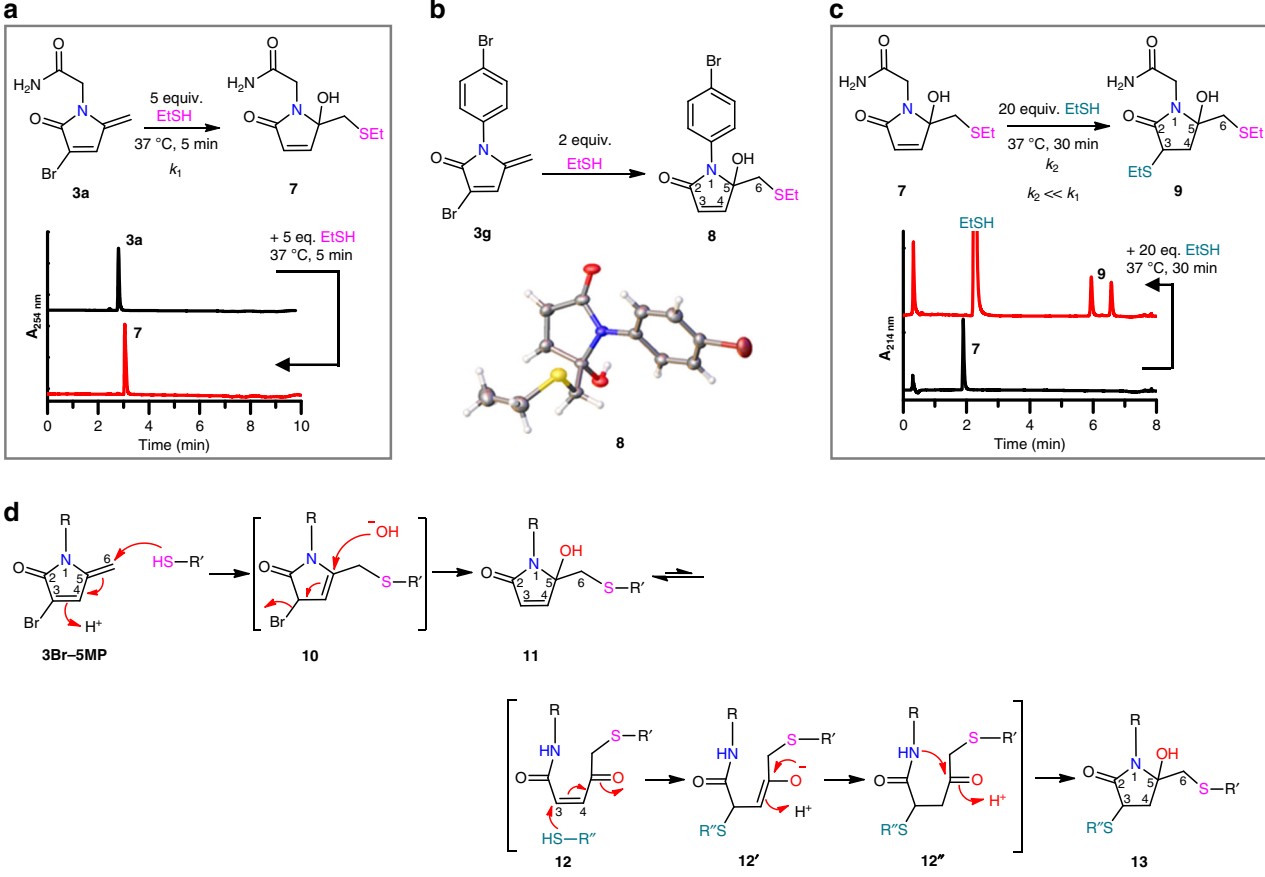

**Fig. 3 Reactivity of 3Br-5MPs with EtSH in pH 7.5 HEPES buffer. a** Reaction of **3a** with EtSH. Lower panel, HPLC analysis of the reaction: Column II, 0–6 min, MeCN from 5 to 30%. **b** Preparation of **8** from **3 g** and EtSH. The crystal structure of **8** is shown below. **c** Reaction of **7** with excess EtSH. Lower panel, HPLC analysis of the reaction: Column I, 0–6 min, MeCN from 5 to 20%. **d** Proposed mechanism for the reaction between 3Br-5MPs with thiols.

3Br-5MPs with the specificities of maleimide and 5MP, we also treated peptide **19** with a large excess of each substrate (10 equiv.) for 1 h. Ultra-performance liquid chromatography–tandem mass spectrometry (UPLC-MS/MS) analyses showed that the use of 3Br-5MP **3a** and 5MP **5** led to modification exclusively on the cysteine, whereas the use of maleimide **6** resulted in a 71% yield of a product that was modified not only on the cysteine residue but also on the N-terminal amino group (Supplementary Fig. 28). That is, the 3Br-5MP showed higher cysteine specificity than maleimide and higher reactivity than 5MP. Moreover, the 3Br-5MP retained its excellent reactivity and specificity over a broad range of pH values (6.0–9.5; Supplementary Fig. 29).

The high reactivity and specificity of the 3Br-5MPs were confirmed by experiments with a histone H3 mutant, H3-V35C (**20**), which contains only one cysteine (at position 35). Reactions of **20** with 2 equiv. of **3a–f** in HEPES buffer (pH 7.5) were complete within 1 h, as indicated by UPLC-MS (Supplementary Fig. 30).

Next, we studied reactions between the mono-adduct **7** and peptide **19** to interrogate the efficiency of further modification of the primary conjugate by cysteine (Table 2). Incubation of peptide **19** with 2 equiv. of **7** at 37 °C for 5 min led to a <1% yield of the conjugated product and the yield increased to 15% when the reaction time was increased to 1 h. When 10 equiv. of **6** was used, the reaction was complete after 1 h and the Cys-modified conjugate was the sole product (Table 2 and Supplementary Figs. 27 and 28). These observations, consistent with the above model studies, indicate that primary conjugate **7** was highly specific for cysteine modification, but its reactivity was

remarkably lower than that of the 3Br-5MPs. These results demonstrate that if the ratio of reagents and the reaction conditions are carefully controlled, 3Br-5MPs can be used not only for cysteine-specific mono-functionalization but also for stepwise multi-functionalization of a single cysteine and disulfide bridging bioconjugation.

**Multi-functionalization of proteins with 3Br-5MPs**. We demonstrated the efficiency of protein multi-functionalization using 3Br-5MPs by tagging H3-V35C (**20**) with both fluorescein and biotin in a stepwise fashion (Fig. 5a). Specifically, reaction of **20** with 2 equiv. of **3e** in HEPES buffer (pH 7.5) at 37 °C for 1 h quantitatively gave fluorescein-modified product **21** as the sole product. Without purification, the reaction mixture was treated with 7 equiv. of biotin-SH (**22**), which was then incubated for 2 h. Doubly labeled product **23** was obtained in yields up to 95% (Supplementary Fig. 31).

To assess the stability of the doubly labeled conjugate, we incubated the purified **23** in HEPES buffer (pH 7.5) at 37 °C. Aliquots were periodically withdrawn and subjected to SDS-polyacrylamide gel electrophoresis analysis. Fluorescence imaging of the gel showed that the fluorescein group was firmly attached on the protein, whereas slow release of the biotin group (via retro-Michael reaction) was observed, as quantified both by immunoblotting with biotin antibody and by MS (Supplementary Fig. 32). Addition of 1 mM glutathione to the incubation buffer markedly increased the rate of biotin release (Fig. 5b) and a glutathione-tagged product **24** was obtained through thiol exchange (Supplementary Fig. 32). Hence, just like the model

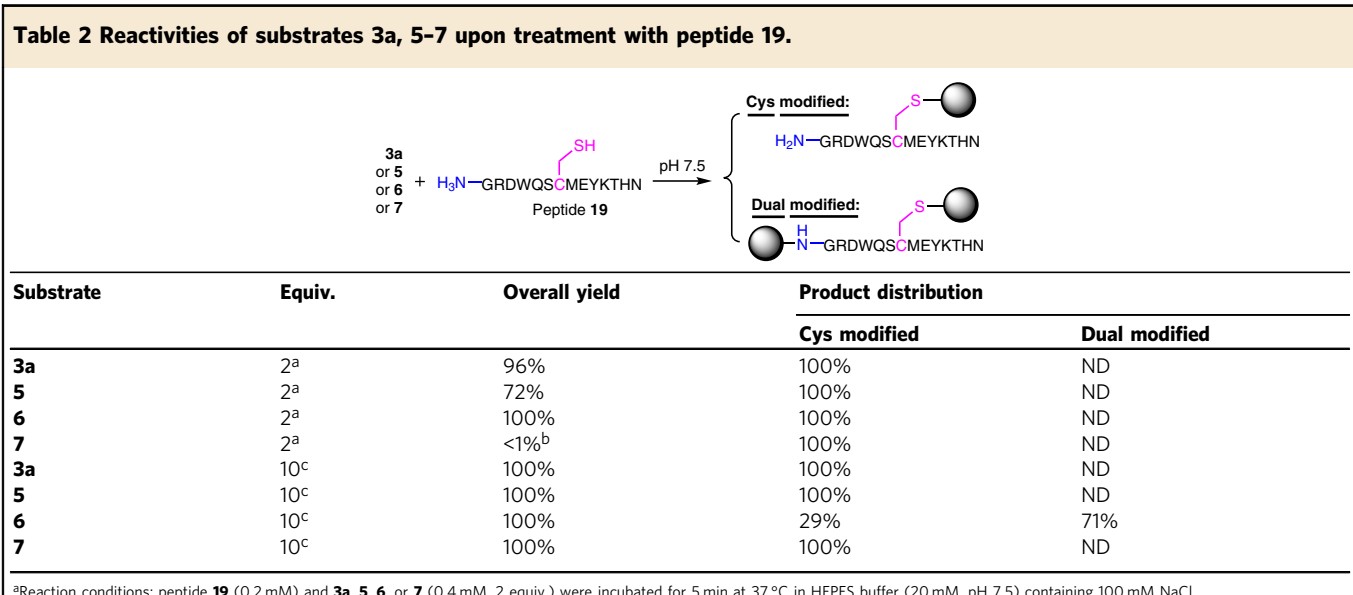

**Fig. 4 Stability and reactivity of EtSH-3Br-5MP adducts 7 and 9. a** HPLC analyses of the thiol-exchange reaction of **9** in the presence of excess BME: Column I, 0–6 min, MeCN from 5 to 50%. **b** Proposed mechanism of thiol-exchange reaction of **9**. **c** Reduction of **9** by NaBH₄ to stabilize the dual-adduct **9**. Lower panel, HPLC analysis of the reduction reaction: Column I, 0–6 min, MeCN from 5 to 50%. **d** Reduction of **7** by NaBH₄ to abolish its further reactivity toward thiols. Lower panel, HPLC analysis of the reduction reaction: Column I, 0–6 min, MeCN from 5 to 20%.

**Table 2 Reactivities of substrates 3a, 5–7 upon treatment with peptide 19.**

| Substrate | Equiv. | Overall yield | Product distribution | |
|---|---|---|---|---|
| | | | Cys modified | Dual modified |
| **3a** | 2[a] | 96% | 100% | ND |
| **5** | 2[a] | 72% | 100% | ND |
| **6** | 2[a] | 100% | 100% | ND |
| **7** | 2[a] | <1%[b] | 100% | ND |
| **3a** | 10[c] | 100% | 100% | ND |
| **5** | 10[c] | 100% | 100% | ND |
| **6** | 10[c] | 100% | 29% | 71% |
| **7** | 10[c] | 100% | 100% | ND |

[a]Reaction conditions: peptide **19** (0.2 mM) and **3a**, **5**, **6**, or **7** (0.4 mM, 2 equiv.) were incubated for 5 min at 37 °C in HEPES buffer (20 mM, pH 7.5) containing 100 mM NaCl.
[b]The overall yield increased to 15% when the reaction time was extended to 1 h.
[c]Reaction conditions: peptide **19** (0.2 mM) and substrate (2 mM, 10 equiv.) were incubated for 1 h at 37 °C in HEPES buffer (20 mM, pH 7.5) containing 100 mM NaCl. Overall yields and product distributions were obtained by integration of ultra-performance liquid chromatography peaks. ND, not detected.

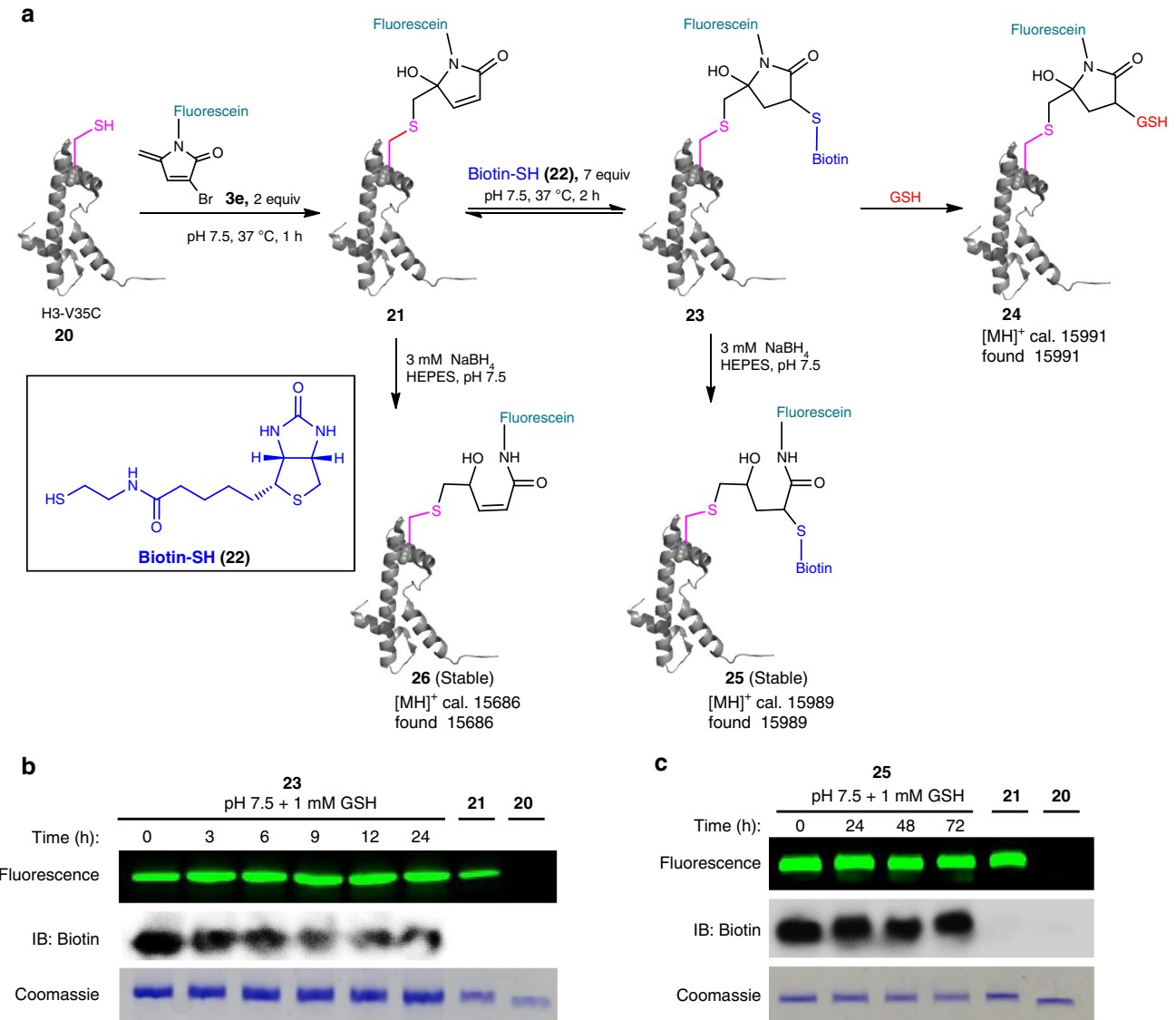

**Fig. 5 Mono- and multi-functionalization of proteins using 3Br-5MPs. a** A schematic representation of mono- and multi-functionalization of H3-V35C (**20**). **b** Fifteen percent SDS-PAGE analysis of the stability of the doubly modified product **23** upon incubation with 1 mM GSH in HEPES buffers (pH 7.5). **c** Fifteen percent SDS-PAGE analysis of the stability of the reduced doubly modified product **25** upon incubation with 1 mM GSH in HEPES buffers (pH 7.5). Source data are provided as a Source Data file.

compound **9**, the dual-functionalized protein conjugate **23** was liable to loss of the secondary modification via thiol-exchange reaction.

We have shown that reduction of **9** by NaBH$_4$ can transform it into stable dual-adducts. In neutral buffers, NaBH$_4$ is a mild reductant and is used for selective reduction of disulfide bonds in proteins[41–43]. We treated the protein H3-V35C (**20**) with different concentrations of NaBH$_4$ in HEPES buffer (pH 7.5) and found that NaBH$_4$ concentration as low as 3 mM rendered no adverse effects for the protein H3-V35C (Supplementary Fig. 33). Thus, **23** was treated with 3 mM (30 equiv.) NaBH$_4$ in HEPES buffer (pH 7.5) and reduced conjugate **25** was obtained (Supplementary Fig. 34). Compound **25** was stable for up to 3 days in a neutral buffer, even in the presence of glutathione (Fig. 5c). The tagging of both fluorescein and biotin on the Cys35 was confirmed at this stage based on MS/MS mapping of the tryptic fragments of **25** (Supplementary Fig. 35). In addition, treatment of mono-functionalized conjugate **21** with 3 mM NaBH$_4$ yielded **26** (Supplementary Fig. 36), which was stable

and not reactive toward biotin-SH (Supplementary Fig. 37). Hence, reduction of protein-3Br-5MP mono-adducts with NaBH$_4$ abolishes their further reactivity toward thiols. This represents an efficient approach to prepare stable and mono-functionalized protein conjugates.

**Disulfide bridging bioconjugation using 3Br-5MPs**. We demonstrated the feasibility of disulfide bridging bioconjugation with 3Br-5MPs by using the endogenous hormone somatostatin (SST)[29,31], which contains a single disulfide bridge. Reduction of the disulfide bond of SST (0.05 mM) with 1.5 equiv. of tris(2-carboxyethyl)phosphine (TCEP) in HEPES buffer (pH 7.5) and subsequent addition of 1.3 equiv. of 3Br-5MP **3a** or **3e** led to disulfide-rebridging product **27** or **28**, respectively, in quantitative yield (Fig. 6a and Supplementary Fig. 38). It is worth noting that the obtained rebidging products are a mixture of regioisomers, depending on which cysteine reacted first with 3Br-5MP. Both **27** and **28** could be converted to the corresponding reduced product

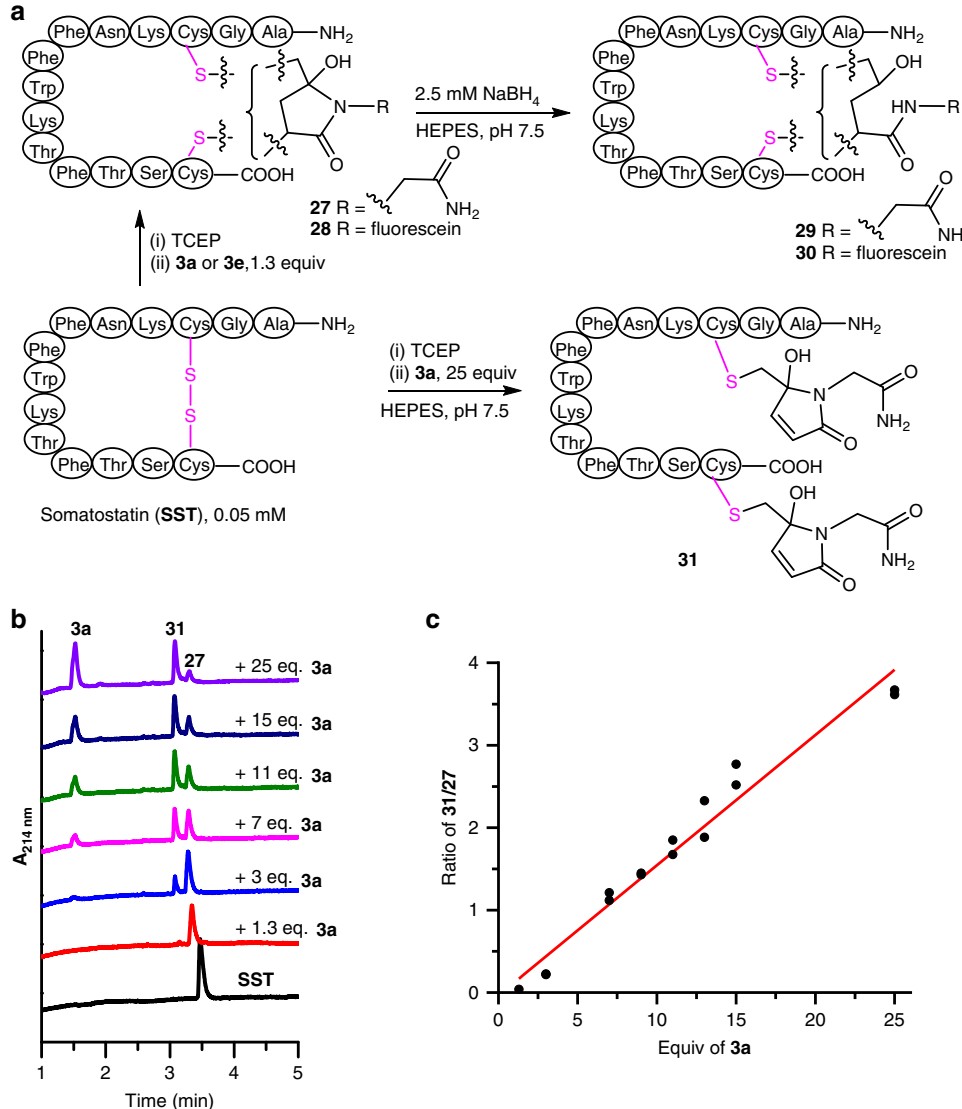

**Fig. 6 Disulfide bridging bioconjugation of somatostatin with 3Br-5MPs. a** Synthesis of somatostatin (SST) bridged by 3Br-5MPs. **b** UPLC analysis of the reactions between reduced SST and various amounts of **3a**. **c** Relationship between **31/27** ratio and amount of **3a**. Data points from two independent experiments are shown here. Source data are provided as a Source Data file.

(**29** or **30**, respectively) by direct addition of 2.5 mM (50 equiv.) NaBH$_4$. Purified **29** and **30** were subjected to Ellman's test, which showed that they contained no free thiol groups, thus confirming that the 3Br-5MPs were attached to SST via disulfide bridging (Supplementary Fig. 38).

In the presence of a relatively small amount of **3a** (<1.3 equiv.), disulfide bridging bioconjugation of SST was efficient and doubly modified product **31** was not detected. However, increasing the amount of **3a** decreased the yield of **27** and led to the formation of **31** (Fig. 6b and Supplementary Fig. 39), the yield of which increased as the amount of **3a** was increased. The **31/27** ratio was linearly correlated with the amount of **3a** (Fig. 6c). These results indicate that after disulfide reduction, 3Br-5MPs could be used not only for bridging bioconjugation but also for modification of two free cysteines and the product selectivity could be easily controlled by adjusting the amount of the 3Br-5MP.

Next, we demonstrated the practicability of preparing antibody conjugates by disulfide bridging with a 3Br-5MP and the antigen-binding fragment (Fab) of goat antibody anti-human IgG (Fig. 7a). The Fab fragment contains two chains that are covalently linked by one interchain disulfide bond. Treatment

of the goat Fab fragment (0.5 mg/mL) in phosphate buffer (20 mM, pH 8.2) with dithiothreitol (0.56 mM, 40 equiv.) led to chemoselective cleavage of the interchain disulfide (Fig. 7b, lane 3)[44,45]. After removal of excess dithiothreitol by gel filtration, 3Br-5MP **3e** (0.02 mM, 1.4 equiv.) was added and the reaction mixture was incubated at 37 °C for 1 h to generate fluorescein-modified Fab fragment **32**. The modified fragment migrated to the same place as the intact fragment (Fig. 7b, lane 5) and the degree of labeling of this conjugate was determined to be approximately 1.1 based on UV-Vis spectrum (Supplementary Fig. 40), suggesting that the two chains were covalently linked by disulfide rebridging via **3e**. In situ reduction by NaBH$_4$ (3 mM) converted **32** to stable conjugate **33**. Given that intrachain disulfides are also reducible by NaBH$_4$[42,43], we carried out a mild oxidation step using dehydrogenated ascorbic acid (0.14 mM)[45] to restore the intrachain disulfide bond (Fig. 7b, lane 7).

To evaluate the biological activity of antibody conjugate **33**, we treated cancer cell line SK-BR-3, which expresses high levels of HER2 on the cell surface, with HER2 antibody human IgG trastuzumab and then with **33**[46]. Fluorescence imaging of the treated cells showed strong fluorescence on the cell surface

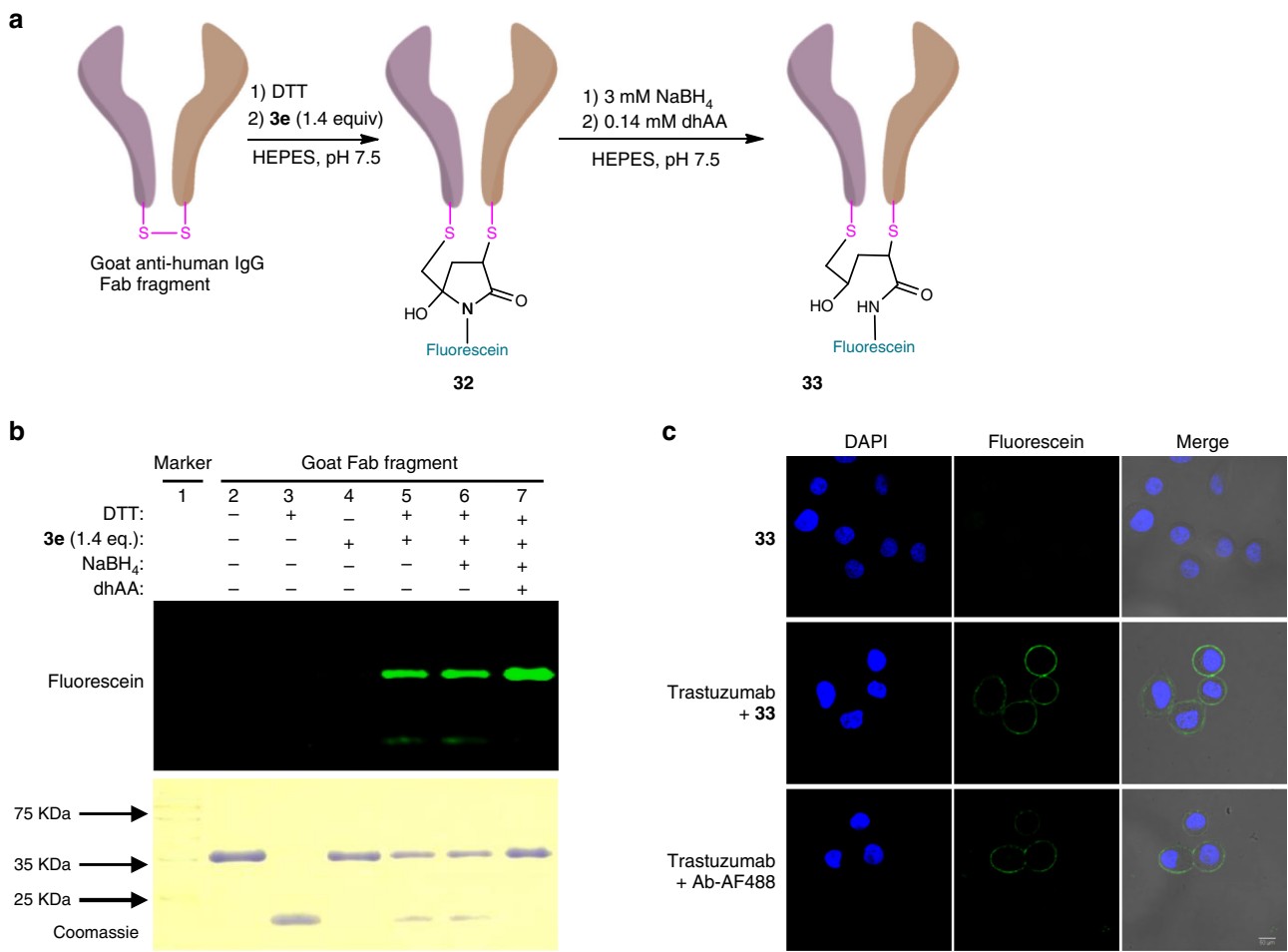

**Fig. 7 Disulfide bridging bioconjugation of an antibody fragment with 3Br-5MP 3e. a** Strategy for disulfide bridging bioconjugation of the Fab fragment of goat anti-human IgG via 3Br-5MP. **b** SDS-PAGE gel analysis of disulfide bridging bioconjugation of the Fab fragment, with fluorescence imaging (top gels) and Coomassie staining (bottom gels). **c** Biological activity of bridged bioconjugate **33**. Trastuzumab binding to HER2 on the surface of SK-BR-3 cells was immunodetected with **33** (second row). Goat anti-human antibody-Alexa488 conjugate (Ab-AF488) at an equivalent concentration was used as a positive control (third row) and an experiment in the absence of trastuzumab was carried out as a negative control (first row). Scale bar, 50 μm.

(Fig. 7c, second row), just as was the case for a positive control experiment in which commercial goat anti-human antibody-Alexa488 conjugate (Ab-AF488) was used as the second antibody. These results indicate that protein modification with the 3Br-5MP, along with reduction with NaBH₄ and oxidation with dehydrogenated ascorbic acid, was mild enough to produce active conjugates for biological applications.

## Discussion

In this study, we demonstrated that 3Br-5MPs, which are analogs of 5MPs and maleimides, nevertheless exhibit different cysteine-specific protein bioconjugation behavior. 3Br-5MPs combine the merits of both 5MPs and maleimides, including easy preparation, good stability, high reactivity, and excellent cysteine specificity. Addition of cysteine to 3Br-5MPs generates conjugates that are amenable to secondary addition by another thiol or cysteine, making 3Br-5MPs valuable for multi-functionalization of a single cysteine and for disulfide bridging bioconjugation. Strikingly, the secondary addition is considerably slower than the primary one, which allows for the controlled preparation of mono-, multi-functionalized proteins or disulfide-bridged conjugates via adjusting the ratio of reagents and reaction conditions. Therefore, we contend that 3Br-5MPs are simple, robust, and versatile reagents for cysteine-specific bioconjugation.

The conjugates obtained by reactions with 3Br-5MPs can be reduced to generate stable products. The labeling reaction and subsequent reduction are mild enough to produce active conjugates for biological applications. Research on preparation of homogeneous antibody–drug conjugates with an ideal drug-to-antibody ratio via disulfide bridging bioconjugation using 3Br-5MPs and elucidation of their pharmacokinetic and pharmacodynamic properties is ongoing in our laboratory.

## Methods

**General procedure for one-pot synthesis of 3Br-5MPs (Method 2)**. To a solution of **4** (100.0 mg, 0.46 mmol, 1.0 equiv.) in 10.0 mL of solvent (5.0 mL tetrahydrofuran (THF) + 5.0 mL of 0.25 M sodium phosphate buffer pH 7.5), N-bromosuccinimide (98.0 mg, 0.55 mmol, 1.2 equiv.) was added. After the solution was stirred for 1 h at 0 °C, amine substrate **2** (0.55 mmol, 1.2 equiv.) was added and the reaction was stirred for 4 h at room temperature. The reaction mixture was concentrated and then subjected to purification by flash chromatography or high-performance liquid chromatography.

**Preparation of dual-functionalized protein conjugate (25)**. A mixture of **3e** (200 μM) and H3-V35C (100 μM) in HEPES buffer (20 mM, pH 7.5, 100 mM NaCl) was incubated at 37 °C for 1 h. Without purification, Biotin-SH (**22**, 700 μM) was added to the solution mixture directly followed by incubation at 37 °C for 2 h. Freshly prepared NaBH₄ solution (100 mM in water) was added directly into the reaction mixture (final concentration 3 mM) and incubated at 37 °C for 40 min. Then excess small molecules and salts were removed by PD Minitrap G-25 column (GE Healthcare).

**Disulfide bridging bioconjugation of SST using 3Br-5MP**. A mixture of TCEP (0.075 mM, 1.5 equiv.) and hormone SST (0.05 mM) in HEPES buffer (20 mM, pH 7.5, 100 mM NaCl) was incubated at 37 °C for 2 h. Without purification, **3a** or **3e** (0.065 mM, 1.3 equiv.) were added into the reaction mixture followed by incubation at the same temperature for 1 h. Freshly prepared NaBH$_4$ solution (100 mM in water) was added (final concentration 3 mM) followed by incubation for another 40 min. Aliquots were taken and analyzed by UPLC-MS.

**Reporting summary**. Further information on research design is available in the Nature Research Reporting Summary linked to this article.

## Data availability

Crystal structure for compound **8** is available as Supplementary Data 1 and also at the Cambridge Crystallographic Data Centre under accession number CCDC: 1961778. Methods and all relevant data are available in Supplementary Information and from the authors. The source data underlying Figs. 5b, 5c, and 6c, and Supplementary Figs. 32a and 37a are provided as a Source Data file.

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

## Acknowledgements

This work was supported by the National Natural Science Foundation of China (21877064 and 91953115 to C.Z.Z. and 21740002 to Z.X.) and by the National Key R&D Program of China (2017YFD0200501 to Z.X.).

## Author contributions

Y.Q.Z. performed most of the biochemical experiments regarding protein modifications. C.L.Z. designed and carried out the chemical synthesis and mechanism studies. G.C.A., M.D.S., and Z.H.C. took part in the chemical synthesis and biochemical experiments. G.C. and Z.X. contributed new reagents/analytic tools. C.Z.Z. designed and supervised

the project. Y.Q.Z., C.L.Z., and C.Z.Z. wrote the manuscript. All authors have given approval to the final version of the manuscript.

## Competing interests

The authors declare no competing interests.
