## [Peer Review File · Nature Communications]

Reviewers' comments:

Reviewer #1 (Remarks to the Author):

The work by Chuanzheng Zhou and co-workers demonstrates analogues of methylene pyrrolones (5MPs) and maleimides, 3-bromo-5-methylene pyrrolones (3Br-5MPs), to target cysteine residues on peptides and proteins. 3Br-5MPs are stable compounds (pH 6, 7.5 and 9.6 for 5 days) and can be prepared in 2-3 steps from easily accessible reagents, offering new candidates for protein modification with good reactivity. Addition of cysteine to 3Br-5MPs generates conjugates that can undergo a second Michael addition by another thiol or cysteine, making 3Br-5MPs interesting compounds for multi-functionalization of a single cysteine and for disulfide bridging ligation. However, the dual-adduct is liable to thiol exchange reaction and in order to abolish its further reactivity toward thiols, the compound needs to be treated with 30 equiv of NaBH₄ for reduction. The same for the conjugates obtained by reactions with 3Br-5MPs that should be reduced to generate stable products. From the results in the paper is clear that the reagent can be successfully applied for peptide/protein modification and disulphide bridging and has superior stability compared to 5MPs and also maleimides after reduction. The need for further NaBH₄ treatment is a major limitation and limits the overall applicability of the method to generate therapeutic conjugates systems. There is no demonstration of the effect of any of the modifications introduced and the conditions used on the structure and functions of the proteins they modify.

Overall the current discoveries reported are interesting from the chemical perspective but lack the overall novelty and applicability to be of interest to the general readership of Nature Communications. I suggest that this work is published in a more specialised journal such as Communications Chemistry.

Other issues:

- None of the LC chromatogram traces and ion series/deconvolution were presented in the SI for the proteins/antibodies employed.
- The overall yield was calculated by integral of LC chromatogram using TIC A280 nm, but this data wasn't shown for any of the reactions.
- The stability of protein H3-V35C upon treatment with NaBH₄ was analysed by LC-MS only. Circular dichroism spectroscopy must be provided to access the overall tertiary structure of the protein and support the stability claim under the applied conditions.
- How to follow if the reduction with NaBH₄ was complete in the protein/antibody reaction? Is it always complete? Can it generate more heterogeneity?
- There are some NMR spectra with significant amount of impurities, e.g. ¹H NMR spectrum of compound 6 and 3g (Figure S9, page S25 and Figure S10, page S26).
- S42, figure S26. The reaction chromatogram shows a broad peak at 1 min that appears to be two coeluting peaks. Please double check. And what is the 1- and 2-min peaks at the BME control?
- S43, Figure 27C: Is the maleimide reacting with the aspartic acid?
- S43, Figure 27 caption: please correct peptide 19 and not 8
- S48, Figure S30: please include a larger mass range (for example 10000-50000 m/z) and also the calculated mass for the conjugates. 3a, 3b and 3f are missing -22 in the mass.
- S49, Figure S31; S51, Figure S33 and S52, Figure S34: please provide the HPLC trace and the ion series before deconvolution. Include larger mass range.
- S50, Figure S32 caption: remove one of the at. "taken at at"

Reviewer #2 (Remarks to the Author):

In this manuscript, the authors describe a new bioconjugation platform, 3-bromo-5-methylene pyrrolones (3Br-5MP). These molecules stem from the development of 5-MP by the same group in 2017 (JACS, 2017, 139, 6146). Like 5-MP, 3Br-5MPs allow the fast, selective and reversible

labelling of cysteines. The introduction of an additional Br atom also allows the sequential addition of a second thiol derivative, thus allowing the double labelling of a single cysteine residue. 3Br-5MPs can also be used for the rebridging of disulfides.

The paper is well constructed and the conclusions are supported by solid experimental studies, with the exception of some points that will be discussed later.

The implementation of the technique appears to be reasonably accessible, suggesting that it could be applied with relative ease. However, other platforms (allyl sulfones, chlorotetrazines) allow the construction of the same type of bioconjugates, which limits the innovative nature of the study.

Some revisions may be made to the manuscript:

The synthesis of 3Br-5MPs is well described. It seems to tolerate the introduction of a wide variety of probes of interest (biotin, fluorophore, cytotoxic drug, etc). The proposed reaction mechanism with thiols seems reasonable and is well supported by structural data.

1) The addition of the first thiol is described as a "Michael addition". This terminology may not be the most appropriate. Michael's addition is traditionally referred to as the 1,4-addition of a nucleophilic carbanion to an alpha,beta-unsaturated carbonyl. The definition has been broadened with the introduction of the "thia-Michael addition" to describe the 1,4-addition of a thiol, but I don't know to what extent this can be used for a 1,6-addition.

In order to prevent thiol exchange and thus stabilize the bioconjugate, the authors use NaBH₄ to reduce the open form of the molecule. This approach is original and effective. However, it can also lead to the reduction of disulfide bridges present in the protein, as observed by the authors. It can also probably lead to the reduction of some imaging probes such as near-infrared fluorophores, which are highly conjugated.

2) Have other, possibly softer, reducing agents been tested? It would be interesting to discuss it.

After validation of the technique on a model peptide, the authors modify a small protein, H3-V35C, on which they site-specifically introduce a fluorescein and a biotin, with satisfactory protein recovery yields.

3) The method of calculation of protein recovery yields should be clarified. In particular, does it take into account the absorbance of fluorescein at 280 nm? (The contribution of fluorescein can be quite important when one measures A₂₈₀ for small proteins). The molar extinction coefficient of the protein H3-V35C could be specified.

In Figure 5, a cyclic peptide (somatostatin) is used as a model for studying the rebridging approach. The study is well conducted.

3) However, as shown in the HPLC chromatograms in Figure S38, the "purified 29 and 30" compounds are actually a mixture of regioisomers (and diastereoisomers), depending on which cysteine reacted first with 3Br-5M. It would be better to include this information in the text or in Figure 5

4) In the same experiment, the disulfide bridge is reduced with TCEP for 2 hours before the addition of 3Br-5MP. Is there a reason why the reduction and conjugation cannot be done simultaneously? Are 3Br-5MPs degraded by TCEP?

Finally, the authors validate their approach with the disulfide rebridging of an antibody fragment.

5) These bioconjugates should be better characterized: mass spectrum + UV-Vis spectrum of the conjugate. This would facilitate the assessment of the degree of labeling and homogeneity of the sample.

Reviewer #3 (Remarks to the Author):

A broad range of fields from biochemistry to cell biology and medicine rely on post-translation modification of proteins to generate conjugates for detection, isolation and various therapies. This is a mature field and much effort has yielded a plethora of reagents for selective derivatization of various protein residues and most particularly Cys. The thiol of this residue is the most nucleophilic site in proteins and the easiest to target. Its history is rich and the bar for innovation is high. The authors of the current submission have made a recent contribution to this field by describing the utility of 5-methylene pyrrolones for modification of Cys residues. This reviewer was very skeptical that a new variation of the pyrrolones could offer much improvement and surpass performance of the current reagents. A careful reading of this manuscript proved otherwise. The newly described 3-bromo-5-methylene pyrrolones demonstrate multiple advantages over existing opportunities. The benefits of these compounds are readily apparent from the clear and thorough characterization presented in this submission. Results are logically described and well illustrated with figures in the text and additional data included in the supplementary materials.

The generality of the authors' strategy is firmly established by the diversity of useful compounds integrated into the protein coupling reagent (Table 1). As summarized in Table 2, the specificity of the 3-bromo derivative for Cys versus the N-terminal α -amine is far superior to that of maleimide, a standard in the field. Additional advantages of the new reagent include its ability to act as a cross-linking reagent with an ability to react with two thiols in a well controlled and sequential manner. This property is successfully revealed in consecutive couplings of a protein to a fluorescent tag and then to biotin (Fig. 4). Another example involves intramolecular cross-linking of two Cys within a peptide hormone somatostatin (Fig. 5) and intermolecular cross-linking between two IgG Fab fragments (Fig. 6). Even better, coupling with the 3-bromo derivative is reversible so that a conjugate may be created for purification or identification and then released to regenerate the native material of interest. Alternatively, reversibility can be halted by a simple treatment of NaBH₄ (Fig. 3).

The versatility of 3-bromo-5-methylene pyrrolones is remarkable and will likely be adopted widely in the field. The detailed description of protocols and results of this manuscript provide a very compelling advance that is worthy of broad dissemination.

Only a few minor revisions are suggested as outlined below:

1. The reaction mechanism summarized in Fig 2D is reasonable but omits a possible enolate intermediate formed after thiol addition and before C3 protonation. Is there evidence for direct 1,4-addition? If so, please describe the data. If not, then include the potential to form the enolate and briefly describe the pros and cons of forming this intervening species.
2. The data establish that compound 11 is not a good Michael acceptor but could this have been anticipated? Can a reasonable rationale be included to explain why it would behave in this manner?
3. The regiochemistry of thiol addition to intermediate 12 is fascinating and may illustrate an important principle that distinguishes between competing Michael additions. Again, chemical rationale for this result should be mentioned when presenting the mechanism of Fig. 2D.

Reviewer #1 (Remarks to the Author):

The work by Chuanzheng Zhou and co-workers demonstrates analogues of methylene pyrrolones (5MPs) and maleimides, 3-bromo-5-methylene pyrrolones (3Br-5MPs), to target cysteine residues on peptides and proteins. 3Br-5MPs are stable compounds (pH 6, 7.5 and 9.6 for 5 days) and can be prepared in 2-3 steps from easily accessible reagents, offering new candidates for protein modification with good reactivity. Addition of cysteine to 3Br-5MPs generates conjugates that can undergo a second Michael addition by another thiol or cysteine, making 3Br-5MPs interesting compounds for multi-functionalization of a single cysteine and for disulfide bridging ligation. However, the dual-adduct is liable to thiol exchange reaction and in order to abolish its further reactivity toward thiols, the compound needs to be treated with 30 equiv of NaBH₄ for reduction. The same for the conjugates obtained by reactions with 3Br-5MPs that should be reduced to generate stable products. From the results in the paper is clear that the reagent can be successfully applied for peptide/protein modification and disulphide bridging and has superior stability compared to 5MPs and also maleimides after reduction. The need for further NaBH₄ treatment is a major limitation and limits the overall applicability of the method to generate therapeutic conjugates systems. There is no demonstration of the effect of any of the modifications introduced and the conditions used on the structure and functions of the proteins they modify.

Overall the current discoveries reported are interesting from the chemical perspective but lack the overall novelty and applicability to be of interest to the general readership of Nature Communications. I suggest that this work is published in a more specialised journal such as Communications Chemistry.

We agree that the need for further NaBH₄ treatment is a limitation. However, we would like to point out that in neutral buffers, NaBH₄ is a mild reductant and is used for selective reduction of disulfide bonds in proteins (Ref. *Bioconjugate Chemistry*, **2014**, 25, 460-469; *Methods in Enzymology* **1987**, 143, 246-256; *Biochimica et Biophysica Acta*, **1960**, 44, 365-367). We have demonstrated that 3 mM NaBH₄ in HEPES buffer (pH 7.5) was sufficient to stabilize the mono- and dual-adduct. Under the same conditions, the structure and function of fluorescein, biotin, α , β -unsaturated amide and the protein itself were not affected, please see Supplementary Fig. 33, 34, 36, 37.

Also, we demonstrated that the reduced Fab-fluorescein conjugate (**33**) showed full biological activity (Fig. 7c), suggesting that protein modification with the 3Br-5MP, along with stabilization with NaBH₄, was mild enough to produce active conjugates for biological applications.

Other issues:

- None of the LC chromatogram traces and ion series/deconvolution were presented in the SI for the proteins/antibodies employed.

Representative LC chromatogram traces and ESI MS series have now been presented in the Supplementary Information. Please see Supplementary Fig. 31, 33, 34 and 36.

- The overall yield was calculated by integral of LC chromatogram using TIC A280 nm, but this data wasn't shown for any of the reactions.

The overall yields of peptide modification were calculated by integral of LC chromatogram (A₂₅₄). The yields are shown in Table 2 and the LC chromatograms are shown in the Supplementary Information, please see Supplementary Fig. 27, 28 and 29. For protein modification, the conjugates could not be separated from the unmodified proteins by LC. Hence, the overall yields of protein modification were calculated based on MS peak intensity. Please see Supplementary Fig. 31, 33, 34 and 36.

- The stability of protein H3-V35C upon treatment with NaBH₄ was analysed by LC-MS only. Circular dichroism spectroscopy must be provided to access the overall tertiary structure of the protein and support the stability claim under the applied conditions.

We appreciate the comment. We have now recorded the CD spectroscopy of H3-V35C upon treatment with NaBH₄. The result confirmed that NaBH₄ treatment does not change the overall tertiary structure of the protein. Please see Supplementary Fig. 33E.

- How to follow if the reduction with NaBH₄ was complete in the protein/antibody reaction? Is it always complete? Can it generate more heterogeneity?

We have analyzed the reduction of model adducts **7** and **9** with NaBH₄ by HPLC (Fig. 4c and d). 1.3 eq. of NaBH₄ led to the complete transformation of **7** and **9** to reduced products in neutral solutions. In addition, HPLC-MS analyses of reduction of SST-disulfide bridging conjugate **27** and **28** with NaBH₄ also revealed that the reduction was very efficient without generating more heterogeneity (Please see Supplementary Fig. 38).

For reduction of protein conjugates, the reduced products could not be separated from the starting materials by HPLC. However, the high-resolution MS revealed that 3 mM of NaBH₄ could drive the reaction to complete in 1 h in pH 7.5 buffer. Please see Supplementary Fig. 34-36.

- There are some NMR spectra with significant amount of impurities, e.g. ¹H NMR spectrum of compound **6** and **3g** (Figure S9, page S25 and Figure S10, page S26).

The NMR spectra contain some solvent peaks. We have now recorded the NMR spectra of compound **6** and **3g** using more pure products. The NMR spectra are proved in the Supplementary Information. Please see Supplementary Fig. 9 and 10.

- S42, figure S26. The reaction chromatogram shows a broad peak at 1 min that appears to be two coeluting peaks. Please double check. And what is the 1- and 2-min peaks at the BME control?

The broad peak seems the oxidized form of BME. We have now carried out the experiment using fresh prepared BME and the broad peak disappeared. Please see Supplementary Fig. 26.

- S43, Figure 27C: Is the maleimide reacting with the aspartic acid?

We thank the reviewer for pointing this out. The maleimide should be conjugated on the Cys. The structure has now been corrected. Please see Supplementary Fig. 27.

- S43, Figure 27 caption: please correct peptide 19 and not 8

We thank the reviewer for pointing this out. This has now been corrected.

- S48, Figure S30: please include a larger mass range (for example 10000-50000 m/z) and also the calculated mass for the conjugates. 3a, 3b and 3f are missing -22 in the mass.

We have now included the ion series for all protein conjugates. The MS spectra after deconvolution were provided in a larger mass range (10000-50000 m/z). The observed MS values of **3a**, **3b** and **3f** were almost identical to the calculated values. Please see Supplementary Fig. 30.

- S49, Figure S31; S51, Figure S33 and S52, Figure S34: please provide the HPLC trace and the ion series before deconvolution. Include larger mass range.

We have now included the ion series and HPLC traces for protein conjugates. The MS spectra after deconvolution were provided in a larger mass range (10000-50000 m/z). Please see Supplementary Fig. 31, 34 and 36.

- S50, Figure S32 caption: remove one of the at. "taken at at"

We thank the reviewer for pointing this out. This has now been corrected.

Reviewer #2 (Remarks to the Author):

In this manuscript, the authors describe a new bioconjugation platform, 3-bromo-5-methylene pyrrolones (3Br-5MP). These molecules stem from the development of 5-MP by the same group in 2017 (JACS, 2017, 139, 6146). Like 5-MP, 3Br-5MPs allow the fast, selective and reversible labelling of cysteines. The introduction of an additional Br atom also allows the sequential addition of a second thiol derivative, thus allowing the double labelling of a single cysteine residue. 3Br-5MPs can also be used for the rebridging of disulfides.

The paper is well constructed and the conclusions are supported by solid experimental studies, with the exception of some points that will be discussed later.

The implementation of the technique appears to be reasonably accessible, suggesting that it could be applied with relative ease. However, other platforms (allyl sulfones, chlorotetrazines) allow the construction of the same type of bioconjugates, which limits the innovative nature of the study.

Some revisions may be made to the manuscript:

The synthesis of 3Br-5MPs is well described. It seems to tolerate the introduction of a

wide variety of probes of interest (biotin, fluorophore, cytotoxic drug, etc). The proposed reaction mechanism with thiols seems reasonable and is well supported by structural data.

1) The addition of the first thiol is described as a "Michael addition". This terminology may not be the most appropriate. Michael's addition is traditionally referred to as the 1,4-addition of a nucleophilic carbanion to an alpha,beta-unsaturated carbonyl. The definition has been broadened with the introduction of the "thia-Michael addition" to describe the 1,4-addition of a thiol, but I don't know to what extent this can be used for a 1,6-addition.

We appreciate the comment. We agree that Michael addition was misused. Hence, in the revised manuscript, we changed "Michael addition" to "addition" regarding the 1,6-addition.

In order to prevent thiol exchange and thus stabilize the bioconjugate, the authors use NaBH₄ to reduce the open form of the molecule. This approach is original and effective. However, it can also lead to the reduction of disulfide bridges present in the protein, as observed by the authors. It can also probably lead to the reduction of some imaging probes such as near-infrared fluorophores, which are highly conjugated.

2) Have other, possibly softer, reducing agents been tested? It would be interesting to discuss it.

We understood the reviewer's concern. We did try softer reducing agents, such as NaBH(OAc)₃. However, neither model compound **7** nor **9** could be efficiently reduced by the softer reducing agent, please see the following results:

After validation of the technique on a model peptide, the authors modify a small protein, H3-V35C, on which they site-specifically introduce a fluorescein and a biotin, with satisfactory protein recovery yields.

3) The method of calculation of protein recovery yields should be clarified. In particular, does it take into account the absorbance of fluorescein at 280 nm? (The contribution of fluorescein can be quite important when one measures A₂₈₀ for small proteins). The molar extinction coefficient of the protein H3-V35C could be specified.

We have recorded the UV spectrum of fluorescein. The ϵ_{280} of fluorescein was determined to be 14400 M⁻¹cm⁻¹ (Please see Supplementary Fig. 40). This value has

been taken into account when calculating the protein recovery. The ϵ_{280} of protein H3-V35C was calculated to be $4595 \text{ M}^{-1}\text{cm}^{-1}$.

In Figure 5, a cyclic peptide (somatostatin) is used as a model for studying the rebridging approach. The study is well conducted.

3) However, as shown in the HPLC chromatograms in Figure S38, the "purified 29 and 30" compounds are actually a mixture of regioisomers (and diastereoisomers), depending on which cysteine reacted first with 3Br-5M. It would be better to include this information in the text or in Figure 5.

We appreciate the comment. I have now included the information in both the text and Fig. 6a. Please see Page 7, the second paragraph.

4) In the same experiment, the disulfide bridge is reduced with TCEP for 2 hours before the addition of 3Br-5MP. Is there a reason why the reduction and conjugation cannot be done simultaneously? Are 3Br-5MPs degraded by TCEP?

Yes, 3Br-5MPs reacts with TCEP. Please see the following results. Actually, we did try to combine the reduction and conjugation in one step, but the result was not as good as the two-step protocol.

Finally, the authors validate their approach with the disulfide rebridging of an antibody fragment.

5) These bioconjugates should be better characterized: mass spectrum + UV-Vis spectrum of the conjugate. This would facilitate the assessment of the degree of labeling and homogeneity of the sample.

The goat Fab anti-human IgG was purchased from SouthernBiotech (Cat. No. 2041-01). It is a heterogeneous mixture after fragmentation and we could not obtain its exact Mass by ESI mass spectrometry. However, the degree of labeling (DOL) of this conjugate was determined to be approximately 1.1 by UV-Vis, Please see Supplementary Figure 40. This result, together with the SDS PAGE analysis (Fig. 7b), proved that each Fab is modified with 1 fluorescein molecule via Disulfide bridging bioconjugation.

Reviewer #3 (Remarks to the Author):

A broad range of fields from biochemistry to cell biology and medicine rely on post-translation modification of proteins to generate conjugates for detection, isolation and various therapies. This is a mature field and much effort has yielded a plethora of reagents for selective derivatization of various protein residues and most particularly Cys. The thiol of this residue is the most nucleophilic site in proteins and the easiest to target. Its history is rich and the bar for innovation is high. The authors of the current submission have made a recent contribution to this field by describing the utility of 5-methylene pyrrolones for modification of Cys residues. This reviewer was very skeptical that a new variation of the pyrrolones could offer much improvement and surpass performance of the current reagents. A careful reading of this manuscript proved otherwise. The newly described 3-bromo-5-methylene pyrrolones demonstrate multiple advantages over existing opportunities.

The benefits of these compounds are readily apparent from the clear and thorough characterization presented in this submission. Results are logically described and well illustrated with figures in the text and additional data included in the supplementary materials.

The generality of the authors' strategy is firmly established by the diversity of useful compounds integrated into the protein coupling reagent (Table 1). As summarized in Table 2, the specificity of the 3-bromo derivative for Cys versus the N-terminal α -amine is far superior to that of maleimide, a standard in the field. Additional advantages of the new reagent include its ability to act as a cross-linking reagent with an ability to react with two thiols in a well controlled and sequential manner. This property is successfully revealed in consecutive couplings of a protein to a fluorescent tag and then to biotin (Fig. 4). Another example involves intramolecular cross-linking of two Cys within a peptide hormone somatostatin (Fig. 5) and intermolecular cross-linking between two IgG Fab fragments (Fig. 6). Even better, coupling with the 3-bromo derivative is reversible so that a conjugate may be created for purification or identification and then released to regenerate the native material of interest. Alternatively, reversibility can be halted by a simple treatment of NaBH₄ (Fig. 3).

The versatility of 3-bromo-5-methylene pyrrolones is remarkable and will likely be adopted widely in the field. The detailed description of protocols and results of this manuscript provide a very compelling advance that is worthy of broad dissemination. Thank you for the overall very positive comments about this manuscript.

Only a few minor revisions are suggested as outlined below:

1. The reaction mechanism summarized in Fig 2D is reasonable but omits a possible enolate intermediate formed after thiol addition and before C3 protonation. Is there evidence for direct 1,4-addition? If so, please describe the data. If not, then include the potential to form the enolate and briefly describe the pros and cons of forming this

intervening species.

There is no evidence for direct 1,4-addition. We have now included the potential enolate intermediate in the Scheme. Please see Fig. 3d. The text has been changed slightly as well. Please see Page 5, the first paragraph.

2. The data establish that compound 11 is not a good Michael acceptor but could this have been anticipated? Can a reasonable rationale be included to explain why it would behave in this manner?

3. The regiochemistry of thiol addition to intermediate 12 is fascinating and may illustrate an important principle that distinguishes between competing Michael additions. Again, chemical rationale for this result should be mentioned when presenting the mechanism of Fig. 2D.

Question 2 and 3 point to the same concern, so we give a combined response:

It is known that the reactivity of different Michael acceptors toward thiols decreases in the following order: α , β -unsaturated aldehyde \approx α , β -unsaturated ketone $>$ α , β -unsaturated esters \gg α , β -unsaturated amide. α , β -Unsaturated amides are not good Michael acceptors. For instance, only the acrylamide without substituents at β position could react with thiols in the presence of catalysts (Please refer to: *Chem. Res. Toxicol.* **2009**, 22, 742; *Rsc Adv.* **2017**, 7, 43104).

In this context, compound **11** is supposed to be not a good Michael acceptor. Actually, we did synthesize an analogue X and found it was not reactive with EtSH (unpublished results):

Similarly, in compound **12**, addition to the C3 (addition to an α , β -unsaturated ketone) is supposed to be much more favorable than an addition to the C4 (addition to an α , β -unsaturated amide).

In the revised manuscript, we added one sentence to discuss the chemical rationale. Please see Page 5, the first paragraph.

REVIEWERS' COMMENTS:

Reviewer #1 (Remarks to the Author):

The authors have improved the manuscript by clarifying some of the questions raised adding new text, providing additional references and/or adding additional data.

However, there are still concerns regarding some of the data presented (such as the absence of the LC traces) as well as its applicability to therapeutic proteins (this is not shown at all). But most importantly this reviewer has concerns about the novelty of the 3-bromo-5-methylene pyrrolones reagents, which are pretty much analogues of extensively used methylene pyrrolones and maleimides. These issues and the lack of novelty of the reagents together with the need to use a reductant, makes this manuscript not suitable for Nature Communications.

Reviewer #2 (Remarks to the Author):

I would like to thank the authors for addressing my comments in a thorough manner. I am now pleased to recommend acceptance of the paper as it is.

Reviewer #3 (Remarks to the Author):

The authors responded quite thoroughly to all of the reviewers' comments and suggestions. Appropriate changes were made including the addition of significant data to the supplemental information section. Revisions have made this excellent manuscript even stronger than it was when first submitted.

Reviewer #1 (Remarks to the Author):

The authors have improved the manuscript by clarifying some of the questions raised adding new text, providing additional references and/or adding additional data.

However, there are still concerns regarding some of the data presented (such as the absence of the LC traces) as well as its applicability to therapeutic proteins (this is not shown at all). But most importantly this reviewer has concerns about the novelty of the 3-bromo-5-methylene pyrrolones reagents, which are pretty much analogues of extensively used methylene pyrrolones and maleimides. These issues and the lack of novelty of the reagents together with the need to use a reductant, makes this manuscript not suitable for Nature Communications.

We have provided representative LC chromatogram traces in the Supplementary Information. Please see Supplementary Figs. 31, 34 and 36. From these LC chromatogram traces, we can see that modified protein could not be separated from the unmodified ones by LC. Thus, LC chromatogram traces did not provide valuable information. Instead, the overall yields of protein modification were obtained based on MS peak intensity. We have provided ion series/deconvolution MS spectra of all modified proteins in the Supplementary Information.

Regarding its applicability to therapeutic proteins, we have demonstrated the practicability of preparing an antibody (Fab)-Fluorescein conjugate by disulfide bridging with a 3Br-5MP, and the prepared Fab-Fluorescein conjugate exhibit full biological activity. Please see Fig. 7. This simple case highlighted the potential applications of 3Br-5MPs in the preparation of therapeutic antibody-drug conjugates. Preparation of homogeneous antibody–drug conjugates via disulfide bridging bioconjugation using 3Br-5MPs and elucidation of their pharmacokinetic and pharmacodynamic properties is another challenging project and will be done in the future.

We do not agree with the referee's comments regarding the novelty of the 3Br-5MP. Though 3Br-5MPs are pretty much analogues of extensively used methylene pyrrolones and maleimides, they exhibit different chemical reactivity and demonstrate multiple advantages. First, 3Br-5MPs show higher cysteine specificity than maleimides and higher reactivity than methylene pyrrolones; Second, 3Br-5MPs can be used for multifunctionalization of a single Cys residue and disulfide bridging bioconjugation, whereas methylene pyrrolones and maleimides can not be used for such purposes.

Reviewer #2 (Remarks to the Author):

I would like to thank the authors for addressing my comments in a thorough manner. I am now pleased to recommend acceptance of the paper as it is.

Thank you for the time and expertise that you provided to improve the manuscript.

Reviewer #3 (Remarks to the Author):

The authors responded quite thoroughly to all of the reviewers' comments and suggestions. Appropriate changes were made including the addition of significant data to the supplemental information section. Revisions have made this excellent manuscript even stronger than it was when first submitted.

We appreciate the comment. We are grateful for the time and expertise that you provided to improve the manuscript.